# Deductive Verification Method of Real-Time Safety Properties for Embedded Assembly Programs

**Satoshi Yamane** 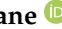

Graduate School of Natural Science and Technology, Kanazawa University, Ishikawa 920-1192, Japan; syamane@is.t.kanazawa-u.ac.jp; Tel.: +81-76-234-4856

**Abstract:** It is important to verify both the correctness and real-time properties of embedded systems. However, as practical computer programs are represented by infinite state transition systems, specifying and verifying a computer program is difficult. Real-time properties are also important for embedded programs, but verifying the real-time properties of an embedded program is difficult. In this paper, we focus on verifying an embedded assembly program, in order to verify the real-time safety properties. We propose a deductive verification method to verify real-time safety properties, based on discrete time, as follows: (1) First, we construct a timed computational model including the execution time from the assembly program. We can specify an infinite state transition system including the execution time of the timed computational model. (2) Next, we verify whether a timed computational model satisfies RTLTL (Real-Time Linear Temporal Logic) formulas by deductive verification. We can specify real-time properties by RTLTL. By our proposed methods, we are able to achieve verification of the real-time safety properties of an embedded program.

**Keywords:** embedded assembly program; verifying real-time safety properties; timed computational model; deductive verification

---

## 1. Introduction

Conventional formal verification is mainly applied to computer hardware and communication protocols. The specifications of these systems are easy to describe, using finite state transition systems. On the other hand, as a practical computer program is represented by an infinite state transition system, specifying and verifying a computer program is difficult. As large-scale computer hardware, such as GPUs and supercomputers, for verifying systems has recently become cheap and as the progress of both abstraction technologies and theorem proof technologies have been remarkable, program verification has also become feasible [1]. Conventionally, verifying embedded systems is important, and embedded program verification is thus also important. Furthermore, real-time properties are important in an embedded program, but verifying the real-time properties of an embedded program is difficult. In this paper, we propose a formal verification method of the real-time safety properties of an embedded assembly program using deductive verification, as follows:

1. First, we construct a timed computational model including the execution time from the assembly program. We can specify an infinite state transition system including the execution time by the timed computational model.
2. Next, we verify whether a timed computational model satisfies RTLTL (Real-Time Linear Temporal Logic) formulas by deductive verification. We can specify real-time properties by RTLTL.

Using our proposed methods, we were able to achieve verification of the real-time safety properties of the embedded program.

This paper is an extension of the previous work in [2]. We emphasize the new original contributions in this paper, as follows:

1. We have implemented our proposed axiom on the theorem prover Princess [3].
2. We have demonstrated experiments with real examples, such as the Linetrace program written for the Wheel-type robot nuvo WHEEL controlled by a H8/3687 microcontroller [4]. This robot is very old, but it has the important features of embedded software.

Henzinger, Manna, and Pnueli pointed out, in their famous paper, that two important classes of real-time requirements for embedded systems are bounded response properties and bounded invariance properties, specified using RTLTL (Real-Time Linear Temporal Logic) [5]. We can check the correctness of systems for all input data by formal verification. On the other hand, we can only check the correctness of systems for some input data by testing. For this reason, in this paper, our approach is formal verification. Additionally, in order to correctly compute the execution times of systems, we verify an assembly program [6].

### 1.1. Outline of This Paper

When we develop embedded software, we may first specify it using hybrid automata or timed automata, and then verify it using model checking. Next, we implement it using C program, and verify it using software model checking. In this paper, in order to verify the real-time properties, including hardware-dependent information, we deductively verify the real-time safety properties of an embedded assembly program, as shown in Figure 1.

The embedded software is described in the C language, but the hardware-dependent part is described as an assembly program. Assembly programs are suitable for a prediction of the execution time. Moreover, the syntax of assembly programs are simple, and their analysis is easy. Therefore, in this paper, we verify an assembly program, instead of a C program.

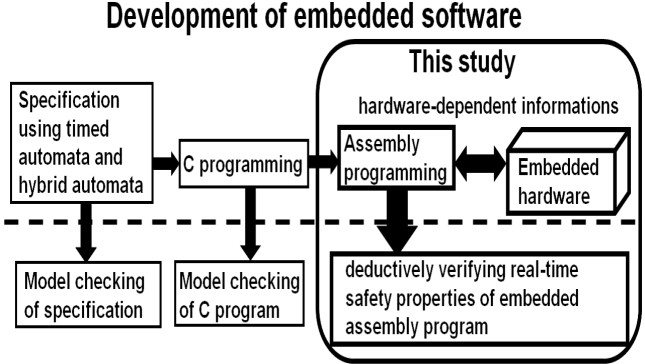

**Figure 1.** Overview of this study.

The advantages of verification of assembly programs as pointed out by Schlich [7] are as follows:

1. The assembly code is the outcome at the end of the development process. Hence, all errors introduced during the complete development process can possibly be found. These errors include errors not visible in intermediate representations (e.g., re-entrance errors), compiler errors, postcompilation errors (e.g., errors introduced by instrumentation code), and hardware-dependent errors (e.g., stack overflows, arithmetic overflows, interrupt handling errors, and writing reserved registers).
2. Assembly language usually has a clean and well-documented semantics. Vendors of microcontrollers provide documentation describing the semantics of the provided assembly constructs. This makes

assembly constructs easier to handle than certain C constructs, such as pointer arithmetic or function calls by pointers.

3. When model checking assembly code, the model checker does not have to exploit the compiler behavior, hardware-dependent constructs can be handled, and the source code (C code) of the software is not required. Hence, even programs that use libraries not available in the source code can be analyzed.

4. Programs consisting of components written in different programming languages can be verified. When model-checking the source code, only single components can be verified and, for each programming language used, a specific model checker has to be utilized.

As shown in Figure 2, we first encode the assembly program into our proposed timed computational model. Secondly, we propose the deductive verification method for real-time safety properties using SMT (Satisfiable Modulo Theories) [8]. Deductive verification method consists of verification rules; one verification rule consists of a temporal formula derived from the premises of first-order formulas. According to [9], we compactly explain a deductive verification method as follows: When we verify whether an assembly program satisfies a safety property, we first encode the assembly program into a timed computational mode. Then, we derive first-order formulas from the timed computational mode, according to a verification rule. Finally, we check validity of the first-order formulas using an SMT solver. If all first-order formulas are valid, the safety property is satisfied.

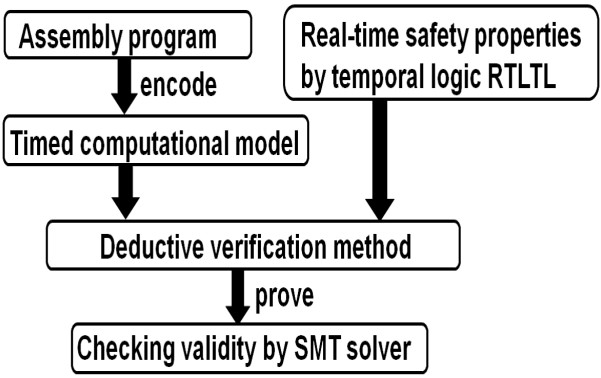

**Figure 2.** Outline of this study.

We propose a timed computational model by assigning the execution time of each instruction to each state.

Using the timed computational model, we can verify the real-time properties of an assembly program. To our knowledge, using a timed computational model that assigns the execution time of each instruction to each state is the first effort in order to verify the real-time properties of an assembly program.

This study is intended for general embedded software, and any RTOS (Real-Time Operating System) is good. We have only adopted H8 as an example In addition, this study is not intended for JAVA programs, as a JAVA program is not an embedded program.

In Section 4, we verify the bounded invariance properties. This example is simple, but we can not verify it by existing methods. Verifying properties related to registers and execution time has been enabled for the first time through our proposed method. We limit this paper to suggesting verification techniques, and more complicated examples are entrusted to future work.

### 1.2. Related Work

In this section, we present related work regarding assembly program verification and formal verification.

1.  Execution time of program

    (a) As both logical correctness and real-time properties are important in embedded systems, a large number of studies regarding the execution time, such as estimating WCET (Worst Case Execution Time), have been explored by researchers [10]. However, in general, due to the behavior of the components which influence the execution time (such as memory, caches, pipelines, and branch prediction), the predicted execution time from program analysis becomes slightly longer than the real execution time. Therefore, it is important and meaningful to verify whether a formula always holds true within a certain time. As the certain time becomes slightly longer than the real execution time, the formula holds true within a real execution time if it holds true within the certain time. This fact makes verifying liveness properties difficult. In this paper, we verify the real-time properties of an assembly program, based on program analysis.

2.  Model checking of program

    (a) Model checkers of C programs using abstraction and refinement methodologies, such as SLAM, BLAST, and MAGIC, have been explored by a large number of researchers [11–13]. However, they have not explored formal verification of the real-time properties.

    (b) Schlich's [mc]square is famous for the study of the verification of assembly programs [7,14]. The [mc]square system utilizes ELF format execution code information, related C language implementation code, static analysis (CFG) of the target assembly code, specification description using CTL, and a model checker. However, [mc]square cannot verify real-time properties. On the other hand, our study can verify real-time properties using RTLTL. We compute timing information using the execution times of assembly instructions. To the best of our knowledge, our paper is the first study of verifying the timing properties of programs.

    (c) The importance of the verification of real-time properties has been pointed out in the model checking of a timed automaton [15]. Campos and Clarke have explored symbolic model checking of discrete real-time systems using RTCTL [16]. All transitions of their timed transition graph happened in one time unit, but the times of timed automata and discrete real-time systems are specified by virtual clocks. However, their model is quite different from our model. A study considering the verification of program execution time has not been carried out, so far, to our knowledge.

3.  Verification of real-time properties of specification

    (a) A. Emerson has explored model checking of discrete real-time systems using RTCTL [17]. On the other hand, Henzinger, Manna, and Pnueli have explored a deductive verification methodology of discrete real-time systems using RTLTL [5]; however, they did not explore real-time verification of real-time programs.

*1.3. Theoretical Background of Program Verification Problem*

In general, program verification problems are theoretically undecidable [18]. In short, no algorithms exist for program verification. However, this problem is partially decidable. If the answer to the problem is "yes", the algorithm will eventually halt with a "yes" answer; if the answer is "no", the algorithm may supply no answer at all. In meaningful cases of real program verification problems, then, the algorithm will eventually halt with a "yes" answer [11–13].

On the other hand, in this paper, we give a deductive temporal verification system based on a Hoare-style axiom system for deductively verifying assembly programs. Cook proved a relatively complete Hoare-style axiom system for program verification [19]; in other words, he proved the relative completeness of a Hoare-style axiom system. Furthermore, Manna and Pnueli have proved a relatively complete proof system for proving the validity of temporal properties of reactive programs [20].

Our proof rule is the extension of Manna's proof system. If we add SMT (Satisfiability Modulo Theories) [8] into our proof rule, we can completely verify assembly programs.

These facts are based on Gödel's Incompleteness Theorem [21].

## 2. Computational Model of Embedded Assembly Program

### 2.1. Embedded Hardware

We show the register set of a H8/3687 processor [22] in Figure 3.

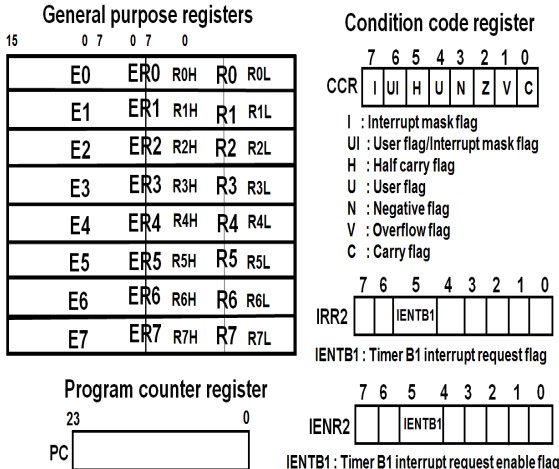

**Figure 3.** Register set of a H8/3687 processor.

In a H8/3687 processor, all the general purpose registers are 32 bits wide. However, the registers can be treated as the concatenation of two 16-bit registers, such as E0 and R0. The 16-bit registers can also be treated as the concatenation of two 8-bit registers, such as RH0 and RL0. On the other hand, control registers consist of a PC (Program Counter), CCR (Condition Code Register), IRR2 (Interrupt Request Register 2), and IENR2 (Interrupt ENable Register 2). In IRR2, when timer B1 overflows, IENTB1 is set to 1. In IENR2, when IENTB1 is set to 1, the overflow interrupt request of timer B1 is admitted.

### 2.2. Computational Model

We propose a timed computational model by assigning the execution time of each instruction to each state. The timed computational model is defined as follows:

**Definition 1.** *(Timed computational model) The timed computational model $C = (V, S, T, \Theta, TM, LAB, time)$ intended to represent an assembly program is given by the following components:*

1. *$V = \{u_0, \ldots, u_{n-1}\}$ is a finite set of variables. The set $V$ consists of program variables, a location, the execution time, registers, and a stack.*
2. *$S$ is an infinite set of states. Each state $s \in S$ assigns to each variable $u_i \in V$ $(i = 0, \ldots, n-1)$ a value.*
3. *$T$ is a finite set of transitions. Each transition $\tau \in T$ is a function $\tau : S \to 2^S$, which can also be represented by a first-order formula $\rho_\tau(V, V\prime)$. Here, the variables in $V$ are the present state variables, and the variables in $V\prime$ are the next state variables.*
4. *$\Theta$ is a satisfiable assertion characterizing all the initial states.*
5. *$TM : S \to N$ is a function assigning to each state $s \in S$ the execution time of the natural number. TM is determined by the hardware manual [22], where the execution time of each instruction is described.*
6. *$LAB : S \to Label$ is a function assigning to each state $s \in S$ an instruction label $\in Label$, where Label is a set of assembly instructions. If no instruction is executed in a state, label is omitted in the state.*

7.　*time represents the total of execution time. The total of the execution time time can be defined for a point in the execution of the model.*

Due to the behavior of the components (such as memory, caches, pipelines, and branch prediction) influencing the execution time, the execution time in this paper becomes slightly longer than the real execution time. However, it is meaningful, from the safety point of view, to verify whether a certain property always holds true within a certain time.

In Section 2.3, we will explain a timed computational model with an example.

### 2.3. Encoding from Assembly Program to Timed Computational Model

The encoding from a program to a state transition system has a standard hand-operated technique [23]; in particular, we refer to pages 14–16 in [23]. Let $V$ be the set of variables. We think of the variables in $V$ as the present state variables and the variables in $V\prime$ as the next state variables. We define a state $s$ to be an interpretation of $V$, assigning to each variable $v$ a value $s[v]$. Furthermore, we denote a state $s$ to be an interpretation of $V$ as the present state variables, and a state $s\prime$ to be an interpretation of $V\prime$ as the next state variables. We denote by $S$ the set of all states, and by $T$ the finite set of transitions. Each transition $\tau \in T$ is a function $\tau : S \to 2^S$, which is also represented by a first-order formula $\rho_\tau(V, V\prime)$.

In this paper, we add both the function $TM(s)$ and a variable time into the set $V$ of variables, where $TM(s)$ expresses the execution time of the assembly instruction in state s, and a variable *time* expresses the total execution time from the initial state.

**Definition 2.** *(Deriving Timed computational model) We show how to derive a timed computational model $C = (V, S, T, \Theta, TM, LAB, time)$ from the first-order formula $\rho_\tau(V, V\prime)$ that represents an assembly program:*

1.　*The set of states S is the set of all valuations for V.*
2.　*Let s and s\prime be two states. Then, $\rho_\tau(V, V\prime)$ holds true when each $v \in V$ is assigned the value $s[v]$ and each $v\prime \in V\prime$ is assigned the value $s\prime[v]$.*
3.　*$\Theta$ is defined by an assertion characterizing all the initial states.*
4.　*TM is determined by the hardware manual [22], in which the execution time of each instruction is described.*
5.　*LAB is defined by a function assigning to each state $s \in S$ an instruction label $\in Label$, where Label is a set of assembly instructions.*
6.　*time is defined by the total execution time from the initial state.*

We show a simple example of part of a timed computational model from an assembly program, as follows.

**Example 1.** *(Example of a timed computational model) An example of a timed computational model generated from assembly program is shown in Figure 4. Each state is defined by the values of variables, registers, individual execution times, stack, and total execution time. As shown in Figure 4, according to [23] and other famous papers, when we derive and specify a state transition system, we omit the \prime representing the next state; in particular, we refer to pages 24–26 in [23].*

*Here, we define $C = (V, S, T, \Theta, TM, LAB, time)$, as in Figure 4, as follows:*

1.　$V = \{Stack, CCR.I, CCR.N, CCR.Z, IRR2flag,$

　　$IENR2flag, @\_c\_start, @\_c\_time, ER0, E1,$

　　$R1, ER5, E6, \_time\}.$
2.　$S = \{s_0, s_1, \ldots\}.$

　　*For example, we describe $s_0$ and $s_1$ as follows:*

　　$s_0 = (Stack = \{\} \wedge CCR.I = 1 \wedge CCR.N = 0$

$\wedge CCR.Z = 0 \wedge IRR2flag = 00111111 \wedge IENR2flag = 00111111 \wedge @\_c\_start = 00000001 \wedge$
$@\_c\_time = 0000000000000000000000000000000 \wedge ER0 = 0000000000000000000000000000000 \wedge$
$E1 = 0000000000000000$
$\wedge R1 = 0000000000000000$
$\wedge ER5 = 0000000000000000000000000000000$
$\wedge E6 = 0000000000000000$

$\wedge\_timer = 11110100000010110 \wedge time = 0).$

$s_1 = (Stack = \{E6\} \wedge CCR.I = 1 \wedge CCR.N = 0$

$\wedge CCR.Z = 0 \wedge IRR2flag = 00111111 \wedge IENR2flag = 00111111 \wedge @\_c\_start = 00000001 \wedge$
$@\_c\_time = 0000000000000000000000000000000 \wedge ER0 = 0000000000000000000000000000000 \wedge$
$E1 = 0000000000000000$
$\wedge R1 = 0000000000000000$
$\wedge ER5 = 0000000000000000000000000000000$
$\wedge E6 = 0000000000000000$
$\wedge\_timer = 11110100000010110 \wedge time = 6).$

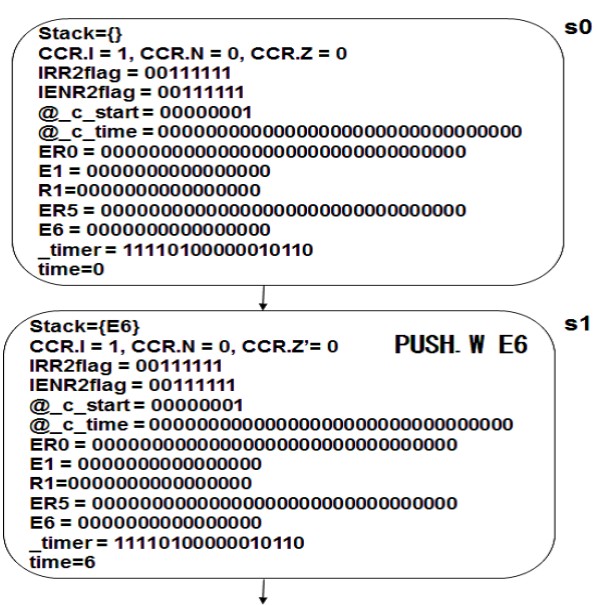

**Figure 4.** Example of part of a timed computational model from an assembly program.

3.  $T = \{\tau_1, \tau_2, \ldots\}.$

    *For example, we describe $\tau_1$, from $s_0$ to $s_1$, as follows.*

    $\tau_1$ *is represented by a first-order formula $\rho_\tau(V, V\prime)$:*

    *(Stack* $= \{\} \wedge CCR.I = 1 \wedge CCR.N = 0$

    $\wedge CCR.Z = 0 \wedge IRR2flag = 00111111 \wedge IENR2flag = 00111111 \wedge @\_c\_start = 00000001 \wedge$
    $@\_c\_time = 0000000000000000000000000000000 \wedge ER0 = 0000000000000000000000000000000 \wedge$
    $E1 = 0000000000000000$
    $\wedge R1 = 0000000000000000$
    $\wedge ER5 = 0000000000000000000000000000000$
    $\wedge E6 = 0000000000000000$
    $\wedge\_timer = 11110100000010110 \wedge time = 0)$

    $\wedge$

    *(Stack$\prime$* $= \{E6\} \wedge CCR.I\prime = 1 \wedge CCR.N\prime = 0$

$\land CCR.Z\prime = 0 \land IRR2flag\prime = 00111111 \land IENR2flag\prime = 00111111 \land @\_c\_start\prime = 00000001$
$\land @\_c\_time\prime \quad = \quad 0000000000000000000000000000000 \quad \land \quad ER0\prime \quad =$
$00000000000000000000000000000000 \land E1\prime = 0000000000000000$
$\land R1\prime = 0000000000000000$
$\land ER5\prime = 00000000000000000000000000000000$
$\land E6\prime = 0000000000000000$
$\land \_timer\prime = 1111010000010110 \land time\prime = 6).$

4. *We assume $\Theta = s_0$.*
5. *We define $TM(s_0) = 0$ and $TM(s_1) = 6$.*
6. *$s_0$ has no label, and $s_1$ has a label (such as PUSH.W E6).*
7. *time is equal to 6 at $s_1$.*

## 3. Deductive Verification Using RTLTL

### 3.1. Real-Time Linear Time Temporal Logic RTLTL

In 1991, Henzinger, Manna, and Pnueli explored RTLTL [5]. RTLTL formulas are constructed from state formulas by Boolean connectives and time-bounded temporal operators.

**Definition 3.** *(Syntax of RTLTL) We inductively define LTL formulae, as follows:*

1. *Each atomic proposition AP is an LTL formula. In this paper, atomic propositions are propositions such as registers, stack, and execution time. For example, the value of a register is 6.*
2. *If p and q are LTL formulae, $p \land q$ and $\neg p$ are LTL formulae.*
3. *If p and q LTL formulae, $pUq$ and $\bigcirc p$ are LTL formulae,*

*where $\bigcirc p$ holds at the current step iff p holds at the next moment, and $pUq$ asserts that q does eventually hold and that p will hold everywhere prior to q.*

*The temporal connective $\Diamond p$ abbreviates $trueUp$ and $\Box p$ abbreviates $\neg \Diamond \neg p$.*

*The temporal connectives $\bigcirc$, $U$, $\Diamond$, and $\Box$ of LTL are extended by timing constraints, and the temporal connectives $\bigcirc_{\leq TIME}$, $U_{\leq TIME}$, $\Diamond_{\leq TIME}$, and $\Box_{\leq TIME}$ of RTLTL are defined, where $TIME$ denotes the constant of execution time.*

*Next, we define bounded invariance and bounded response properties. In this paper, we focus on bounded invariance.*

1. *Bounded invariance:*

   $\Box_{\leq TIME} q, \Box(p \rightarrow \Box_{\leq TIME} q),$
2. *Bounded response:*

   $\Diamond_{\leq TIME} q, \Box(p \rightarrow \Diamond_{\leq TIME} q).$

### 3.2. Deductive Verification Using RTLTL

In this paper, we extend the deductive verification method explored by Manna and Pnueli [9]. As for the axiom of the deductive verification of temporal logic, the temporal logic formula is derived by premises to consist of predicate logic formulas [9] . This is the most important work of Amir Pnueli, which won the ACM A.M. Turing Award in 1996 [24]. This study expands on A. Pnueli's study by adding execution time, and develops the axiom of deduction verification using RTLTL.

In this paper, as shown in Figure 5, we construct a part of our verification axiom $\Box_{\leq TIME} q$ as $\Box(q \land (time \leq TIME))$ over a timed computational model. In Figure 5, we introduce a variable *time* to measure execution time.

In consideration of Figure 5, we define the verification axiom $\Box_{\leq TIME} q$, as shown in Figure 6.

In Figure 6, if Premises B1 and B2 are valid, $\Box_{\leq TIME} q$ is obviously valid. Therefore, this axiom is sound. Furthermore, our timed transition model is the same as Henzinger's timed transition system [5]

when the minimal delay is equal to the maximal delay. Therefore, we can prove that our verification axiom is relatively complete by Henzinger's proof technique [5].

When we verify an assembly program using our verification axiom, a set of first-order formulae are constructed. We can verify whether each formula is valid or not using an SMT solver [3].

**Figure 5.** Part of the verification axiom of $\square_{\leq TIME}\, q$.

**Figure 6.** Verification axiom of $\square_{\leq TIME}\, q$.

Premise B1 requires that the time is set to $TM(s_0)$ at an initial state and the initial condition $\Theta$ implies $(q \wedge (time \leq TIME))$. Premise B2 requires that the time is set to $TM(s_i) + time$ and all transitions preserve $(q \wedge (time \leq TIME))$.

## 4. Experiments of Deductive Verification of Real-Time Properties

We try to deductively verify embedded an assembly program. We used the Linetrace program written, for the Wheel-type robot nuvo WHEEL controlled by a H8/3687 microcontroller [4]. The Linetrace program acquires values from a sensor, and operates a robot from the values. The robot has three sensors and a motor: the sensors can distinguish black from white, and output either 0 or 1 by color; the motor is controlled by PID control. When a timer overflow interrupt of timer B1 occurs, H8/3687 acquires the value from a sensor, and sets the new current targeted value from the value. When a timer overflow interrupt of timer V occurs, H8/3687 performs PID control from the current targeted value and the current value, and outputs the value in the motor.

In this section, we verify a timer interrupt function *_int_tim_b*1. If *_int_tim_b*1 is executed, it acquires the value of the sensor and decides the current targeted value. It returns to processing before the interrupt.

An assembly program of a timer interrupt function *_int_tim_b*1 is shown in Figure 7.

```
_int_tim_b1:      ; function: int_tim_b1
      PUSH.W    E6
      PUSH.W    R5
      PUSH.L    ER1
      PUSH.L    ER0
      BCLR.B    #5,@65527:8
      BCLR.B    #5,@65525:8
      MOV.B     @_c_start:16,R0L
      BEQ       L111:8
      JSR       @_control_linetrace:16
      JSR       @_control_abort:16
      MOV.L     @_c_time:16,ER0
      ADD.L     #10,ER0
      MOV.L     ER0,@_c_time:16
L111:
      SUB.W     E1,E1
      SUB.W     E6,E6
L112:
      MOV.W     E6,R1
      MOV.L     @(_timer:16,ER1),ER0
      BEQ       L113:8
      DEC.L     #1,ER0
      MOV.L     ER0,@(_timer:16,ER1)
      BRA       L114:8
L113:
      MOV.W     #LWORD _timer,R5
      ADD.W     R1,R5
      MOV.W     #1,R0
      MOV.W     R0,@(8:16,ER5)
      MOV.L     @(4:16,ER5),ER0
      MOV.L     ER0,@ER5
L114:
      INC.W     #1,E1
      ADD.W     #10,E6
      CMP.W     #2,E1
      BLT       L112:8
      BSET.B    #5,@65525:8
      POP.L     ER0
      POP.L     ER1
      POP.W     R5
      POP.W     E6
      RTE
```

**Figure 7.** Assembly program of a timer interrupt function *_int_tim_b*1.

We show a timed computational model of a timer interrupt function *_int_tim_b*1 in Figure 8. In Figure 8, we describe only the values that have changed from the previous state in the current state.

1. First, a state is defined by the values of stacks, flags, variables, timers, and execution times. Execution time is the number of states, and one state is 0.05 microseconds.
2. Next, we verify whether $\square_{\leq 75} \, (E1 = R1)$ holds true. When $\square_{\leq 75} \, (E1 = R1)$ holds true, we also check whether the program has reached an error state (*time* = 76 and $E1 = R1$).

The statement $\square_{\leq 75} \, (E1 = R1)$ means that, within execution time 75, the registers E1 and R1 are equal. As existing tools can not compute the execution time, we can not verify $\square_{\leq 75}(E1 = R1)$ using existing tools [7].

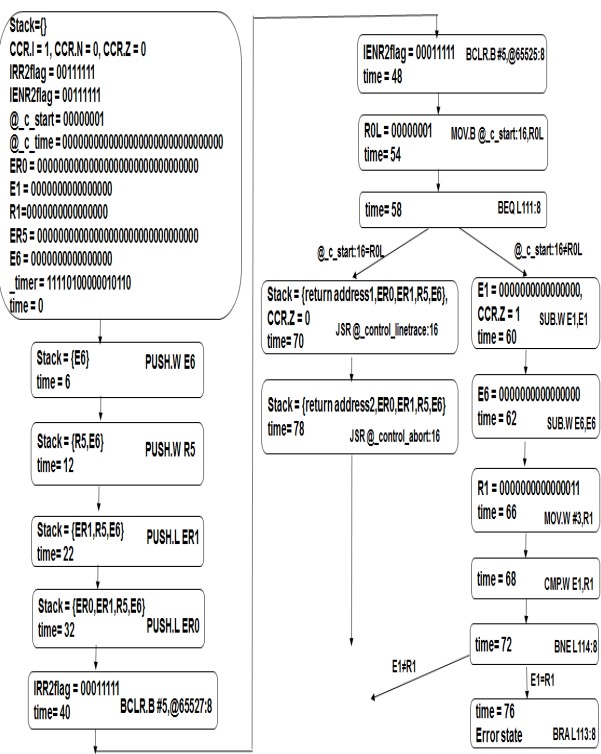

**Figure 8.** A timed computational model of _int_tim_b1.

3.  Next, we verify whether $\square_{\leq 75}\,(E1 = R1)$ holds true, as follows:

(a)  B1. $time = TM(s_0), \Theta \rightarrow q$.

Initially, let $time = 0$, where $time$ measures the total execution time.

$\Theta$ is as follows:

$stack = \{\}$
$\wedge CCR.I = 1$
$\wedge CCR.N = 0$
$\wedge CCR.Z = 0$
$\wedge IRR2flag = 00111111$
$\wedge IENR2flag = 00111111$
$\wedge @\_c\_start = 00000001$
$\wedge @\_c\_time =$
00000000000000000000000000000000
$\wedge ER0 = 00000000000000000000000000000000$
$\wedge E1 = 0000000000000000$
$\wedge R1 = 0000000000000000$
$\wedge ER5 = 00000000000000000000000000000000$
$\wedge E6 = 0000000000000000$
$\wedge \_timer = 1111010000010110$
$\wedge time = 0.$

Let $\square_{\leq 75}\,(E1 = R1) = \square((E1 = R1) \wedge (time \leq 75))$. As $time = 0$ and $E1 = R1$ hold true, $\Theta \rightarrow q$ is valid. We check whether $\Theta \rightarrow q$ is valid using the SAT/SMT solver Princess [3].

(b)  B2. $time = TM(s_i) + time, \{q \wedge ((time - TM(s_i)) \leq TIME)\}T\{q \wedge (time \leq TIME)\}$:

where $\{q \wedge ((time - TM(s_i)) \leq TIME)\}T\{q \wedge (time \leq TIME)\}$ is equal to $\{q \wedge ((time - TM(s_i)) \leq TIME)\} \wedge \rho_\tau \rightarrow \{q\prime \wedge (time\prime \leq TIME)\}$.

- $(E1 = R1) \wedge (time \leq 75) \wedge (Stack = \{\}$
  $\wedge CCR.I\prime = CCR.I$
  $\wedge CCR.N\prime = CCR.N$
  $\wedge CCR.Z\prime = CCR.Z$
  $\wedge IRR2flag\prime = IRR2flag$
  $\wedge IENR2flag\prime = IENR2flag$
  $\wedge @\_c\_start\prime = @\_c\_start$
  $\wedge @\_c\_time\prime = @\_c\_time$
  $\wedge ER0\prime = ER0$
  $\wedge E1\prime = E1$
  $\wedge R1\prime = R1$
  $\wedge ER5\prime = ER5$
  $\wedge E6\prime = E6$
  $\wedge \_timer\prime = \_timer$
  $\wedge time = 0$
  $\wedge Stack\prime = \{E6\}$
  $\wedge time\prime = 6)$
  $\rightarrow$
  $\{(E1\prime = R1\prime) \wedge time\prime \leq 75\}.$

  We checked whether the above first-order formula is valid using the SAT/SMT solver Princess [3], as shown in the Appendix A. The above first-order formula is valid.
- $(E1 = R1) \wedge (time \leq 75) \wedge (Stack = \{ER0, ER1, R5, E6\}$
  $\wedge CCR.I\prime = CCR.I$
  $\wedge CCR.N\prime = CCR.N$
  $\wedge CCR.Z\prime = CCR.Z$
  $\wedge IRR2flag\prime = IRR2flag$
  $\wedge IENR2flag\prime = IENR2flag$
  $\wedge @\_c\_start\prime = @\_c\_start$
  $\wedge @\_c\_time\prime = @\_c\_time$
  $\wedge ER0\prime = ER0$
  $\wedge E1\prime = E1$
  $\wedge R1\prime = R1$
  $\wedge ER5\prime = ER5$
  $\wedge E6\prime = E6$
  $\wedge \_timer\prime = \_timer$
  $\wedge time = 68$
  $\wedge Stack\prime = Stack$
  $\wedge time\prime = 72)$
  $\rightarrow$
  $\{(E1\prime = R1\prime) \wedge time\prime \leq 75\}.$

  We checked whether the above first-order formula is valid using SAT/SMT solver Princess [3]. The above first-order formula is valid.
- $(E1 = R1) \wedge (time \leq 75) \wedge (Stack = \{ER0, ER1, R5, E6\}$
  $\wedge CCR.I\prime = CCR.I$
  $\wedge CCR.N\prime = CCR.N$
  $\wedge CCR.Z\prime = CCR.Z$
  $\wedge IRR2flag\prime = IRR2flag$
  $\wedge IENR2flag\prime = IENR2flag$

$$\land @\_c\_start\prime = @\_c\_start$$
$$\land @\_c\_time\prime = @\_c\_time$$
$$\land ER0\prime = ER0$$
$$\land E1\prime = E1$$
$$\land R1\prime = R1$$
$$\land ER5\prime = ER5$$
$$\land E6\prime = E6$$
$$\land \_timer\prime = \_timer$$
$$\land time = 72$$
$$\land Stack\prime = Stack$$
$$\land time\prime = 76)$$
$$\rightarrow$$
$$\{(E1\prime = R1\prime) \land time\prime \leq 75\}.$$

We checked whether the above first-order formula is valid using the SAT/SMT solver Princess [3]. The above first-order formula is valid.

Finally, when $\Box_{\leq 75} (E1 = R1)$ holds true, the function does not arrive at an error state, in which case $time = 76$ and $E1 = R1$ hold true.

Finally, $\Box_{\leq 75} (E1 = R1)$ holds true.

## 5. Conclusions and Future Work

We have proposed a deductive verification method in order to verify the real-time safety properties of an embedded assembly program, in the following manner:

1. First, we constructed a timed computational model including the execution time from the assembly program. We could specify an infinite state transition system including the execution time by the timed computational model.
2. Next, we developed a verification axiom of $\Box_{\leq TIME} q$. We verified whether a timed computational model satisfies RTLTL (Real-Time Linear Temporal Logic) formulas by deductive verification.
3. We also implemented our proposed axiom in the theorem prover Princess [3].
4. Finally, we demonstrated experiments with real examples, such as the Linetrace program written for the Wheel-type robot nuvo WHEEL controlled by the H8/3687 microcontroller [4]. This robot is very old, but it has the important features of embedded software.

Using our proposed methods, we were able to achieve verification of the real-time safety properties such as $\Box_{\leq TIME} q$ of an embedded assembly program. In addition, we have demonstrated experiments with real examples. It is worth noting that deductive verification is tedious work.

We are now working on the model-checking of real-time safety and liveness properties of assembly program [25]. Furthermore, we are working on a CEGAR-based model checking method [26], based on the work of Rybalchenko [27]. Our recent results [25,26] are fully automatic verification methods without manual verifications, freeing you from tedious work. However, even if we can verify some program by our proposed method in this paper, we can not verify the program by our recent results [25,26]. The best way would be to combine these two approaches.

**Acknowledgments:** This work was supported in part by JSPS/MEXT Grant-in-Aid for Scientific Research Numbers 15K00093.

**Conflicts of Interest:** The authors declare no conflicts of interest.

## Appendix A

We show the method for checking whether the first-order formula mentioned above is valid using the SAT/SMT solver Princess [3], as follows.

We input the program codes shown in Figure A1 into Princess.

```
1    import ap.SimpleAPI
2    import ap.parser.IFormula
3    import ap.SimpleAPI.ProverStatus
4
5    import scala.collection.mutable.ArrayBuffer
6
7    object Princess {
8      def main(args: Array[String]) =
9        SimpleAPI.withProver { prov =>
10         import prov._
11
12         val current : State = new State
13         current.InitState
14
15         val spe : ArrayBuffer[IFormula] = ArrayBuffer()
16
17         current.ccr.I_=("1")
18         current.ccr.N_=("0")
19         current.ccr.Z_=("0")
20         current.irr2.IRR2_=("00111111")
21         current.ienr2.IENR2_=("00111111")
22         current.a_c_start.ACStart_=("00000001")
23         current.a_c_time.ACTime_=("0000000000000000000000000000000000")
24         current.reg.ER_=(0, "000000000000000000000000000000000")
25         current.reg.E_=(1, "0000000000000000")
26         current.reg.R_=(1, "0000000000000000")
27         current.reg.ER_=(5, "000000000000000000000000000000000")
28         current.reg.E_=(6, "0000000000000000")
29         current.a_timer.ATimer_("11110100000010110")
30         current.time.Time_= (0)
31
32         val next : State = new State
33         next.CopyState(current)
34
35         next.stack.Push("E6")
36         next.time.Time_=(next.time.Time + 6)
37
38         spe += current.stack.Stack.isEmpty
39         spe += next.ccr.CCR.equals(current.ccr.CCR)
40         spe += next.irr2.IRR2.equals(current.irr2.IRR2)
41         spe += next.ienr2.IENR2.equals(current.ienr2.IENR2)
42         spe += next.a_c_start.ACStart.equals(current.a_c_start.ACStart)
43         spe += next.a_c_time.ACTime.equals(current.a_c_time.ACTime)
44         spe += next.reg.ER(0).equals(current.reg.ER(0))
45         spe += next.reg.E(1).equals(current.reg.E(1))
46         spe += next.reg.R(1).equals(current.reg.R(1))
47         spe += next.reg.ER(5).equals(current.reg.ER(5))
48         spe += next.reg.E(6).equals(current.reg.E(6))
49         spe += next.a_timer.ATimer.equals(current.a_timer.ATimer)
50         spe += current.time.Time.equals(0)
51         spe += next.stack.Peek.equals("E6")
52         spe += next.time.Time.equals(6)
53
54         for(i <- spe.toList) !!(i)
55
56         if (???.equals(ProverStatus.Sat)) {
57           println("0 is Sat")
58           reset
59           !!(if (next.time.Time<=75) true else false)
60           printf("next run cycle: %d\n", next.time.Time)
61         } else {
62           println("0 is Unsat")
63         }
64       println(???)
```

**Figure A1.** Example of program codes input into Princess.

From lines 17–30, the current state is specified. From lines 38–52, the state transition is specified.

Following this, Princess proves the formula and outputs the verification result shown in Figure A2. Here, Sat is the output. If the quantifier-free formula, such as $stack = \{\} \land CCR.I = 1 \land CCR.N = 0 \ldots$, is satisfiable, the formula is valid. Therefore, the first-order formula is valid.

```
0 is Sat
next run cycle: 6
Sat

Process finished with exit code 0
```

**Figure A2.** Princess output of the verification result.

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
