# Peer review of "Deductive Verification Method of Real-Time Safety Properties for Embedded Assembly Programs"

_electronics, doi:10.3390/electronics8101163_

Round 1
Reviewer 1 Report
Please present the problem you are addressing, the methodology used, and the results achieved in the abstract. There is a typo on line 20 "implemented". Citations 3 and 4 are not accessible. Also, the work cited is very old. Please spell RTLTL, PC, CCR, IRR2, IENR2. Line 131 "Register" R should be small. Figure 5 is repeated. The result section needs to be explained in more detail.
Author Response
We add revised paper and letter by pdf file.

Reviewer 2 Report
The paper proposes the deductive verification method of embedded assembly program for verifying real-time safety properties using RTLTL by generating a timed computational model.
The results are not surprising but may have some interest in the research field.
Hence I suggest to accept the paper.
Author Response
We improve english.

Round 2
Reviewer 1 Report
Even though the paper has improved from its last version but it still lacks a detailed explanation of the result section. It would be interesting to see the experimental setup. Also, it would be nice if you can compare your results with the recently published results. In section 2.2 computational model is described. Please explain it with an example.
Author Response
(1)We added a detailed explanation of the result section.
Also we compared our results with the recently our published results.
(2)In section 2.3 we have already explained a computational model with an example.
